# Transcriptome and DNA Methylome Analysis of Two Contrasting Rice Genotypes under Salt Stress during Germination

**DOI:** 10.3390/ijms24043978

**Published:** 2023-02-16

**Authors:** Yongqiang Li, Dianjing Guo

**Affiliations:** State Key Laboratory of Agrobiotechnology, School of Life Sciences, The Chinese University of Hong Kong, Hong Kong 999077, China

**Keywords:** seed germination, salt stress, DNA methylation, transcriptome, direct seeding, rice

## Abstract

With climate change and labor shortages, direct-seeding rice cultivation is becoming popular worldwide, especially in Asia. Salinity stress negatively affects rice seed germination in the direct-seeding process, and the cultivation of suitable direct-seeding rice varieties under salinity stress is necessary. However, little is known about the underlying mechanism of salt responses during seed germination under salt stress. To investigate the salt tolerance mechanism at the seed germination stage, two contrasting rice genotypes differing in salt tolerance, namely, FL478 (salt-tolerant) and IR29 (salt-sensitive), were used in this study. We observed, that compared to IR29, FL478 appeared to be more tolerant to salt stress with a higher germination rate. GD1 (germination defective 1), which was involved in seed germination by regulating alpha-amylase, was upregulated significantly in the salt-sensitive IR29 strain under salt stress during germination. Transcriptomic data showed that salt-responsive genes tended to be up/downregulated in IR29 but not in FL478. Furthermore, we investigated the epigenetic changes in FL478 and IR29 during germination under saline treatment using whole genome bisulfite DNA sequencing (BS-seq) technology. BS-seq data showed that the global CHH methylation level increased dramatically under salinity stress in both strains, and the hyper CHH differentially methylated regions (DMRs) were predominantly located within the transposable elements regions. Compared with FL478, differentially expressed genes with DMRs in IR29 were mainly related to gene ontology terms such as response to water deprivation, response to salt stress, seed germination, and response to hydrogen peroxide pathways. These results may provide valuable insights into the genetic and epigenetic basis of salt tolerance at the seed germination stage, which is important for direct-seeding rice breeding.

## 1. Introduction

As one of the most important crops worldwide, rice (*Oryza sativa* L.) feeds about half the population around the world [1]. However, the growth, development, and yield of rice are strongly affected by environmental cues. Salinity is a major source of plant abiotic stresses that inhibits plant growth, development, and productivity. The data from the Global map of Salt-affected Soils indicates that more than 833 million hectares of land worldwide are salt-affected [2]. Soil salinity stress causes extensive losses to rice yield annually. Salinity affects rice at any stage during the whole life cycle. Seed germination is a critical stage of plant life that plays essential roles in seed-seedling transition and seedling establishment. It can be inhibited by various environmental factors, such as cold, heat, hypoxia, and salinity. It has become increasingly important to improve rice salt tolerance at the germination stage, as direct seeding is becoming more popular in many countries because it is more cost-effective and easier to use [3]. Thus, it is crucial to comprehend the genetics of salt tolerance during rice germination to establish an efficient breeding system for salt tolerance.

The quantitative trait loci (QTLs) mapping method has been widely used to identify QTLs conferring salt tolerance during germination. Many QTLs were identified in rice at the germination stage under salt stress [4,5]. With the advance of next-generation high throughput sequencing, the transcriptomics approach has been widely used to reveal the transcription changes under salt stress at the seedling stage in Arabidopsis, rice, soybean, sesame, and maize [6,7,8,9,10]. Germination can be affected by endogenous and external factors, e.g., phytohormones and environmental conditions. For instance, gibberellic acid promotes seed germination, and salinity inhibits seed germination. Gibberellic acid and abscisic acid play antagonistic roles in seed germination [11]. Transcriptome and metabolome have been performed in an indica and a japonica variety to reveal the transcriptional changes during germination [12]. Riguang and Xiangxuejing15 were the most salt-tolerant variety and the most salt-sensitive variety, respectively, in the 114 rice varieties germination test [13]. It was reported that seed germination was inhibited by salt stress in the Nipponbare rice variety, and the downregulated alpha-amylase genes under salt stress were related to the inhibited germination [14].

DNA methylation is one of the most important epigenetic modifications that regulates gene expression, transposable element activity, and heterochromatin formation. In plants, DNA methylation occurs in three sequence contexts: CG, CHG, and CHH (H denotes A or T, or C). METHYLTRANSFERASE 1 (MET1) is in charge of CG methylation maintenance, whereas CHROMOMETHYLASE 3 (CMT3) and CHROMOMETHYLASE 2 (CMT2) are for CHG and CHH maintenance, respectively [15]. The RNA-directed DNA methylation (RdDM) pathway is a plant-specific and de novo DNA methylation mechanism [16]. In Arabidopsis, PolIV, PolV, RDR2, DCL3, AGO4/AGO6, and DRM2 are critical components in RdDM [16]. DNA methylation is dynamically regulated by DNA methyltransferases and DNA demethylases. The REPRESSOR of SILENCING 1 (ROS1) family is the major DNA demethylase in plants [17]. Growing evidence shows that DNA methylation responds to biotic and abiotic stresses [18,19]. Salt stress can induce DNA methylation alterations in different rice genotypes [20]. High-affinity potassium transporters (HKTs) are involved in the homeostasis of sodium and potassium in plants under salinity stress [21]. DNA methylation changed the expression of some HKTs in Arabidopsis and wheat under salt stress [22,23]. Two contrasting rice genotypes, FL478 and IR29, showed spatial-temporal DNA methylation changes under salt stress [3]. The salt-tolerant rice genotype Pokkali was more flexible than the salt-sensitive IR29 in DNA demethylation upon salt stress [24]. The salt-induced transcription factors, AtMYB74 and GmMYB84, were regulated by DNA methylation under salinity in Arabidopsis and Soybean, respectively [25,26]. Low-concentration NaCl (25 mM) treatment induced decreased DNA methylation compared to the mock treatment in rapeseeds [27].

Despite all the research efforts described above, little is known about the transcriptomic changes and the epigenetic responses of rice seeds under salt stress during seed germination. In this study, we performed RNA sequencing and whole-genome bisulfite DNA sequencing to investigate transcription and DNA methylation changes in two contrasting rice genotypes during seed germination under salinity stress. Genome-wide CHH hypermethylation was observed under salinity stress during germination, and most CHH hyper DMRs were located within TEs.

## 2. Results

### 2.1. The Effects of Salinity Stress on Seed Germination in Two Contrasting Genotypes

The seedling salt-tolerant rice genotype FL478 and sensitive genotype IR29 seeds were chosen to study the effects of salt stress on seed germination. The phenotype analysis results indicated that germination was inhibited in both genotypes two days after imbibition, whereas for the tolerant genotype FL478, the inhibition was minor compared to the sensitive genotype IR29 (Figure 1A). The radicle and coleoptile were inhibited significantly for the genotype IR29. For the genotype FL478, coleoptile was inhibited, which is comparable to IR29. However, the radicle protruded and was inhibited slightly compared with control. Transcriptome sequencing was performed to study the effects of salt stress on mRNA transcription during germination. PCA analysis indicates that two biological replicates of each sample clustered together, which means good replication quality (Figure 1B). Salt-treated samples were far from the control and explained mainly by PC1. The difference between IR29 and FL478 was explained mainly by PC2, and collectively PC1 and PC2 explained over 94% of the total variation. There were about 18,000 genes expressed (TPM >= ≥ 1) in all samples (Figure 1C). However, the numbers of condition-specific expressed genes were limited in both genotypes (Figure 1D).

### 2.2. Differentially Expressed Genes and GO Enrichment Analysis

In total, 2701 and 3564 differentially expressed genes (DEGs, |log2FC| ≥ 1 & padj < 0.05) were detected in FL478 and IR29, respectively. Among these DEGs, 600 upregulated genes and 847 downregulated genes were shared by FL478 and IR29 (Figure 2A). These DEGs were further divided into FL478-specific, IR29-specific, and shared DEGs. GO enrichment analysis was then performed for the above three DEGs subsets. Some IR29-specific GO terms included response to abscisic acid, response to water deprivation, response to cold, cellular response to hypoxia, response to hydrogen peroxide, and regulation of secondary root formation (Figure 2B). Two GO terms, regulation of growth and plant-type cell wall organization, appeared only in FL478 (Figure 2B). The GO terms response to oxidative stress, cell wall organization, hydrogen peroxide catabolic process, auxin-activated signaling pathway, response to auxin, and metal ion transport appeared in both genotypes (Figure 2B).

### 2.3. Salt Stress Induced GD1 Expression in Salt Sensitive IR29

GD1 (germination defective 1), a B3 domain-containing transcription factor, was reported to affect rice seeds germination [28]. In this study, the expression of GD1 increased significantly in the sensitive IR29 under salt stress, but no change in the tolerant FL478 (Figure 3A). GD1 regulated carbohydrate metabolism by affecting two key starch degradation enzymes, OsAMY1A and OsAMY3C [28]. In our study, we found that OsAMY1A mRNA expression was downregulated significantly in IR29 under salinity stress (Figure 3A). Salinity stress thus induced GD1 expression, which inhibits OsAMY1A in the sensitive genotype IR29 during germination.

### 2.4. The Expression Patterns of Important Salt Responsive Genes

ROS has been reported to regulate seed dormancy, germination, and deterioration [29] by breaking seed dormancy and triggering seed germination at a low level [29]. In the imbibition process, NADPH oxidases are ROS-producing enzymes. In this study, we found that RBOHh (RESPIRATORY BURST OXIDASE HOMOLOG H, or NADPH oxidase 9) was upregulated significantly in IR29 (Figure 3B). We also found that SOD1 and CAT-A were downregulated significantly in the salt-sensitive genotype IR29 (Figure 3B).

Ion homeostasis plays an important role in salt response. We found some ion homeostasis-related genes expressed differentially in the two genotypes. For example, HKT7, a high-affinity K+ transporter, was downregulated significantly in IR29 but remained stable in FL478 (Figure 3C). As the most crucial salt tolerance sensor, SOS1 (SALT OVERLY SENSITIVE 1) mRNA was upregulated in IR29 (Figure 3C). AHA2 (plasma membrane H+−ATPase 2) mRNA was also increased in IR29, but only a slight increase in FL478 (Figure 3C).

We then compared the mRNA expression of salt-responsive genes in the two genotypes. Some salt tolerance and osmotic stress related genes showed different expression patterns in the two genotypes. Among them, PCF5, OsDi19-5, ARAG1, and bZIP71 were downregulated significantly in the salt-sensitive genotype IR29 but only showed slight changes in the salt-tolerant genotype FL478 (Figure 3D). DREB1A, DLN247, OsIQM2, P5CS2, bHLH148, OCPI2, and OsCCD1 were upregulated only in IR29 (Figure 3D). Four of these genes were validated by qRT-PCR (Figure 3E).

### 2.5. Salt-Responsive Transcription Factors during Germination

Transcription factors are essential in the development and abiotic stresses. To reveal the salt-responsive transcription factors in seeds during germination, we extracted those differentially expressed TFs, and in total, 213 and 256 differentially expressed TFs were identified in FL478 and IR29, respectively. These TFs accounted for about 10% of total DEGs (Figure 4A). The two genotypes shared 121 differentially expressed TFs and the expression pattern of these differentially expressed TFs was divided into 4 clusters. Among them, TFs in cluster 2 showed a dramatically decreased expression pattern in IR29 compared with those in FL478 (Figure 4B). TF expression in cluster 3 increased dramatically in IR29 compared with FL478. TFs in cluster 1 showed a dramatically decreased expression pattern in FL478 compared to IR29. In contrast, TFs expression in cluster 4 dramatically increased in FL478 compared to IR29.

The TF family was further analyzed to reveal more detailed information about these differentially expressed TFs. Results showed that MYB, AP2/ERF, bHLH, WRKY, bZIP, and NAC were top-listed enriched transcription factor families (Figure 4C,D). In total, there were 40 and 33 differentially expressed TF families in IR29 and FL478, respectively.

### 2.6. Global CHH DNA Methylation Increase under Salt Stress

To profile the DNA methylation pattern at the genome level, the average DNA methylation level of CG, CHG, and CHH in one-Mb bin was plotted as a heatmap in each sample (Figure 5A). For CG and CHG, the methylation level was significantly higher around the centromeric region, consistent with previous studies that stated DNA methylation mainly occurred in heterochromatin regions. In contrast, CHH methylation was predominately distributed in faraway centromeric regions. The CG and CHG methylation patterns were comparable in the two genotypes, as well as in control vs. saline conditions (Figure 5A). However, the CHH methylation rate increased dramatically in both genotypes under salinity stress compared with the control. To further reveal the differences between control and salt stress in three contexts, kernel density plots of methylation differences in 200-bp windows throughout the genome were plotted (Figure 5B). The peak shift means that the methylation level increased globally in one sample. As shown, both genotypes presented a CHH peak shift between salt and control conditions (Figure 5B). The results indicated that the CHH methylation level increased dramatically under salt stress in both tolerant and sensitive genotypes.

We then analyzed the key DNA demethylases, methyltransferases, and RdDM components. Only genes with TPM greater than 1 in at least one sample were retained. It was found that the DNA methylase OsDML3b was downregulated in both genotypes (Figure SA). Another DNA demethylase OsROS1A was downregulated only in FL478 (Appendix A). For RdDM components, OsRDR2 and OsRDR3 were only upregulated in FL478 (Appendix A). OsAGO4a was only upregulated in IR29 (Appendix A).

### 2.7. High CHH Methylation Level in Gene and TE under Salinity Stress

Previous studies indicated that DNA methylation usually affects gene expression through methylation in the promoter region and deactivates the TE activity by high methylation in the TE body [17]. To study the DNA methylation on different genomic features, we calculated the methylation level of gene promoters and the TE body and compared the difference between salt-treated and control groups in two genotypes. For protein-coding genes, the CG and CHG methylation level of gene promoters increased significantly in IR29 under salt stress (Figure 6A). In contrast, the methylation level was kept stable in FL478 in control and treatment groups (Figure 6A). For TE, the methylation rate in both genotypes and the treatment group did not change in the context of CG (Figure 6B). In the case of CHH, the methylation level increased significantly under saline stress in both genotypes (Figure 6B). For CHG, the methylation rate slightly increased under salt stress compared with the control group for both genotypes. We also used metaplot to profile the DNA methylation level along genes and TEs. For protein-coding genes, it seems that the methylation level at the flank region of the protein-coding gene was higher than the gene body except for in the case of CG (Figure 6C). The comparison between salt stress and control groups indicated that the methylation rate increased significantly in salinity stress, especially in the flank regions, in CHH but not in CG and CHG contexts. A slight increase of CHH methylation under salt stress was observed in IR29 compared with FL478 (Figure 6C). For TEs, the methylation level in the TE body was higher than in the flank regions in all three contexts. The CG methylation level was identical in different genotypes and conditions. The CHH methylation level increased dramatically, especially in TE bodies, in contrast to protein-coding genes.

We also calculated the DNA methylation level in different TE families (Appendix A). Miniature Inverted-repeat Transposable Elements (MITEs), with the highest CHH DNA methylation, increased dramatically under salt stress in both genotypes. Furthermore, the methylation rate of MITE in IR29 was higher than that in FL478 under salinity stress. All these results demonstrate that the CHH methylation level was high in the protein-coding gene and TE in both genotypes under salt stress.

### 2.8. Differentially Methylated Regions under Salt Stress

We identified differentially methylated regions (DMRs) in each context following the criteria by Stroud [30]. The analysis results showed that in the CG context, there were few hyper and hypo DMRs between salt and control groups in the two genotypes (Appendix A). In the CHG context, there were 3552 DMRs (1945 hyper DMRs and 1607 hypo DMRs) in IR29 and 2158 DMRs (1105 hyper DMRs and 1052 hypo DMRs) in FL478 (Appendix A). For CHH, there were a considerable number of hyper DMRs detected in IR29 (71,685) and FL478 (63,692), but only a few hypo DMRs in IR29 (863) and FL478 (2606). These DMRs were then annotated to different genomic features, including upstream 1kb of the gene, downstream 1kb of the gene, gene body, TE, and intergenic regions, to reveal their characteristics. The result showed that about half of the hyper DMRs were in TE regions (Figure 7A).

Due to the vast amount of hyper CHH DMRs, further analysis was focused on these DMRs. We found many genotype-biased hyper CHH DMRs, and only about 30% were shared by both genotypes (Figure 7B). The distribution of genotype-biased and shared hyper CHH DMRs was plotted on 12 chromosomes (Figure 7C).

### 2.9. Differentially Expressed Genes with DMRs

To examine the effects of DMRs on gene expression, we calculated the association between CHH DMRs and the differentially expressed genes. There were 1093 DEGs associated with CHH DMRs in FL478 and 1537 DEGs associated with CHH DMRs in IR29 (Figure 8A). Next, we compared the DEGs with DMRs (DMG) in both genotypes and identified 443 DEGs shared by the two genotypes (Figure 8B). Then, 650 FL478-specific DEGs, 1094 IR29-specific DEGs, and 443 shared DEGs were submitted for GO enrichment analysis. For the biological process GO terms, response to water deprivation, response to salt treatment, response to cold, biosynthetic process, seed germination, response to hydrogen peroxide, and response to hypoxia, appeared only in IR29 (Figure 8C). For GO terms of molecular function, iron ion binding, transmembrane transporter activity, and oxidoreductase activity were more enriched in IR29 (Figure 8C). DNA methylation around genes is often associated with gene silencing. Consistent with the role of DNA methylation in gene silencing, increased DNA methylation was observed for those downregulated DEGs (Figure 8D). We also found that CHH methylation level at promoters was positively correlated with gene expression (Appendix A). For the upregulated DEGs, the increase in DNA methylation corresponded to increased gene expression (Figure 8D).

## 3. Discussion

The molecular basis for rice salt tolerance at the germination stage is less well understood. It has been reported that plants exhibit different sensitivities to salinity at different stages. For example, maize is sensitive to salt at the germination stage but tolerant at the seedling stage [31]. Thus, this research aimed to study the salt response at the germination stage by comparing two contrasting rice genotypes, FL478 and IR29, which were salt-tolerant and salt-sensitive rice genotypes at the seedling stage, respectively. Little is known about the salt response of these two genotypes at the germination stage. In this work, we found that although the germination of both genotypes was inhibited under 150 mM NaCl treatment (Figure 1A), FL478 showed higher tolerance than IR29. We speculate that it may be due to decreased bioactive GA content under salt stress because salt can inhibit rice seed germination by reducing bioactive GA (i.e., GA1 and GA4) activity [14].

Significant transcriptional changes were observed in rice seeds germinated under salt stress, and the identified DEGs were mainly related to abiotic stress (Figure 2B). GO enrichment analysis results of DEGs in FL478 and IR29 showed that response to water deprivation, salt tolerance, and the ion toxicity pathway was highly enriched (Figure 2B). Salinity may inhibit seed germination in two ways: 1. induces high osmotic potential to reduce water uptake and stop nutrient mobilization for germination; 2. salt ions cause toxicity to the embryos [31]. GD1 (germination defective 1) was involved in seed germination and negatively regulated GA and carbohydrate metabolism [28]. In this study, we found that GD1 was only upregulated in the salt-sensitive genotype IR29 (Figure 3A). This suggested that salinity stress enhanced the expression of GD1 and inhibited GA and carbohydrate metabolism during germination. Whether the increased salt tolerance in FL478 is partly due to the stable GD1 expression is worth further investigation. In addition, many salt-responsive genes were only upregulated or downregulated under salinity stress in the sensitive IR29. For example, the dehydration-responsive element-binding protein 1A (DREB1A) and pyrroline-5-carboxylate synthase (P5CS) were upregulated significantly in IR29. DREB1A is a transcriptional activator that binds specifically to the C-repeat/DRE element and mediates cold-inducible transcription. CBF/DREB1 factors play a key role in freezing tolerance and cold acclimation. P5CS is a bifunctional enzyme that exhibits glutamate kinase (GK) and γ-glutamyl phosphate reductase (GPR) activities. In most plant species, two isoforms of pyrroline-5-carboxylate synthetase (P5CS) catalyze the first step in proline biosynthesis from glutamate. P5CS1 has been identified as the major contributor to stress-induced proline accumulation [32]. For ion homeostasis, the expression of HKT7, SOS1, and AHA2 changed significantly in the salt-sensitive genotype under salt stress, but no apparent changes were observed in the salt-tolerant genotype (Figure 3B). As a sodium transporter, HKT7 has been suggested to confer salt tolerance to durum wheat by regulating cellular Na^+^ concentration [33]. AHA2 is a plasma membrane H^+^-ATPase that plays an important role in the plant response to low-phosphorus stress. AHA2-deficient plants exhibited reduced primary root elongation and lower H^+^ efflux in the root elongation zone [34]. The stable AHA2 expression in the salt-tolerant rice genotype FL478 led us to suspect that this gene may contribute to mediating root proton (H+) flux and confer salt tolerance at the germination stage. Further study using gene-deficient mutants may help reveal the detailed function of these candidate genes that differentiated the salt-tolerant and salt-sensitive genotypes.

As an epigenetic marker, DNA methylation regulates many biological processes and responses to all biotic and abiotic stresses. Few studies have investigated the function of DNA methylation in rice seeds during germination under salinity stress. Although salinity stress inhibits seed germination, little is known about the potential role of epigenetic regulation in salt stress during rice seed germination. Comprehensive methylome coupled with transcriptome was used in this study to reveal the potential role of DNA methylation in seed germination under salt treatment. We generated single-base resolution maps of DNA methylation for rice seeds under control and salt treatment two days after imbibition (Figure 5). We found that the DNA methylation level in CG and CHG contexts remained stable under salt stress for both genotypes. In contrast, the CHH methylation level increased dramatically and globally. About half of the hyper CHH DMRs were in TEs (Figure 7A). In rice, TEs prefer to insert into 5’ flanking sequences of genes and render adjacent genes stress-inducible [35]. Activated TEs can induce DNA hypermethylation via RdDM to repress TE-associated genes upon stress [16].

DNA methylation can be dynamically regulated by DNA methylases and demethylases. CHH methylation is mainly maintained by CMT2. We observed that the expression of CMT2 decreased significantly in IR29 under salt stress but only slightly decreased in FL478 (Appendix A). The result is contrary to high CHH methylation. Thus, it is necessary to study the association between CMT2 and salt stress to understand the roles of CMT2 in the future. ROS1 and DML3 are responsible for DNA demethylation in rice and Arabidopsis, and four ROS1 genes (OsROS1a, OsROS1b, OsROS1c, and OsROS1d) and two DML3 genes (OsDML3a and OsDML3b) have been identified in rice [36]. In this study, only the expression of OsDML3b decreased significantly in both genotypes under salt treatment, consistent with the high CHH methylation level (Appendix A). Furthermore, ROS1A was upregulated in FL478 under salt stress. RNA-directed DNA methylation is a plant-specific DNA methylation pathway that involves de novo DNA methylation [16]. The expression of critical genes in RdDM was therefore analyzed in this study. However, except for OsAGO4a, none of the key genes in RdDM were significantly upregulated or downregulated during germination under salinity stress. OsAGO4a was slightly upregulated in IR29 under salt treatment, whereas OsRDR2 and OsRDR3 were upregulated in FL478 (Appendix A). Based on the above results, we inferred that RdDM activity was not highly active in this process. Instead, the demethylation process predominantly decreased during rice germination under salt stress. We speculate that hyper CHH methylation might be caused by increased RdDM and decreased demethylation activity under salt stress during germination. DNA demethylation occurs either through an active or a passive process. It has been shown that the DNA methylation level decreases gradually during germination in Arabidopsis and soybean [37]. Previous evidence showed that Arabidopsis seeds undergo passive global CHH demethylation during germination [38]. Therefore, another possible reason that CHH methylation remained at a high level is that passive dilution of DNA methylation over cell division and replication in the absence of maintenance enzymes is inhibited by high salinity.

## 4. Materials and Methods

### 4.1. Plant Materials and Salt Treatment

Rice indica genotypes, FL478 (salt-tolerant) and IR29 (salt-sensitive), were provided by the International Rice Research Institute. Seeds with the glume removed were sterilized with 20% bleach for ten minutes and washed three times with distilled water. Seeds were germinated on filter paper in a petri-dish under 28 °C in the dark. Control and salt-treated samples were treated with water and 150 mM NaCl, respectively. Samples were collected at two days after imbibition.

### 4.2. RNA-Seq and Data Analysis

Total RNA from FL478 and IR29 was extracted with the RNeasy plant mini kit following the manufacturer’s instructions (QIAGEN; Hilden, Germany). Then total RNA was treated with DNase I (NEB; Ipswich, MA, USA) to remove DNA. Two biological replicates were kept for each treatment. PolyA mRNA was isolated from 4-ug total RNA by the Poly(A) mRNA Magnetic Isolation Module (NEB). Then the library was prepared following the TRACE-seq protocol [39]. Finally, libraries were sent to the Novogene Co. Ltd. (Tianjin, China) for sequencing.

After removing low-quality reads and adaptors, clean reads were mapped to the MH63 genome reference (MH63RS3) by HISAT2 [40] with at most two mismatches. FeatureCounts was used to count mapped reads for each gene [41]. Differentially expressed genes were identified by the R package DESeq2 (v1.26.0) [42] with the following criteria: abs(log2Fold change) ≥ 1 and FDR < 0.05. GO enrichment analysis was completed using Rice Genome Hub and plotted by ggplot2 package.

### 4.3. Bisulfite-Seq and Data Analysis

Genomic DNA was extracted from seeds using the DNeasy plant mini kit (QIAGEN) with RNA removal. 1-ug genomic DNA was sonicated by Covaris M220 (SKLA, CUHK, Hongkong, China). After DNA fragmentation, 300–700-bp fragments were selected by VHTS clean beads (Vazyme; Nanjing, China). Then, selected DNA was conducted end repair, dA-tailing, and methylated adaptor ligation by the NEBNext Ultra II DNA Library Prep Kit according to the manufacturer’s instructions (NEB). Adaptor ligated fragments were then treated with bisulfite using the EZ DNA Methylation-Gold kit (Zymo; Irvine, CA, USA). After PCR amplification, the PCR products were purified by DNA clean beads (Vazyme). Finally, the BS-seq libraries were sent to Novogene for pair-end sequencing.

For data analysis, briefly, paired-end raw reads were trimmed with trim_galore (v0.6.4) and mapped to the MH63 genome reference (MH63RS3) using Bismark (v0.22.3) [43] with default parameters. Deduplicated reads were discarded, and unique mapping reads were retained for further analysis. DMRs were identified by DMRcaller (v1.26.0) [44] with a 100-bp bin, at least 5 cytosines covered for each cytosine, and at least 5 cytosines in the bin. Fisher’s exact test was performed to detect DMRs with an FDR < 0.05 and an absolute methylation difference over 0.4, 0.2, and 0.1 for CG, CHG, and CHH, respectively. Circos plots of DNA methylation at the genome level were plotted by Circos software (v0.69.8) [45]. Kernel density plots were generated by comparing the average methylation rate within a 200-bp window between CK and Salt. Windows with at least 5 cytosines and each covered at least 5 reads in at least one sample were considered.

## 5. Conclusions

In this study, we investigated the influence of salinity stress on rice seed germination by transcriptome and methylome analysis. We found that FL478 was more tolerant to salt stress at the germination stage compared to IR29. It was speculated that upregulated GD1 with downregulated alpha-amylase1A may lead to germination inhibition under salt stress. CHH hypermethylation under salt stress during germination also regulates seed germination and cellular response to salt stress. Future studies will focus on revealing the molecular mechanism of GD1 in response to salt stress during germination. The ROS, SOD, and Na+/K+ level should be studied in the future.

## Figures and Tables

**Figure 1 ijms-24-03978-f001:**
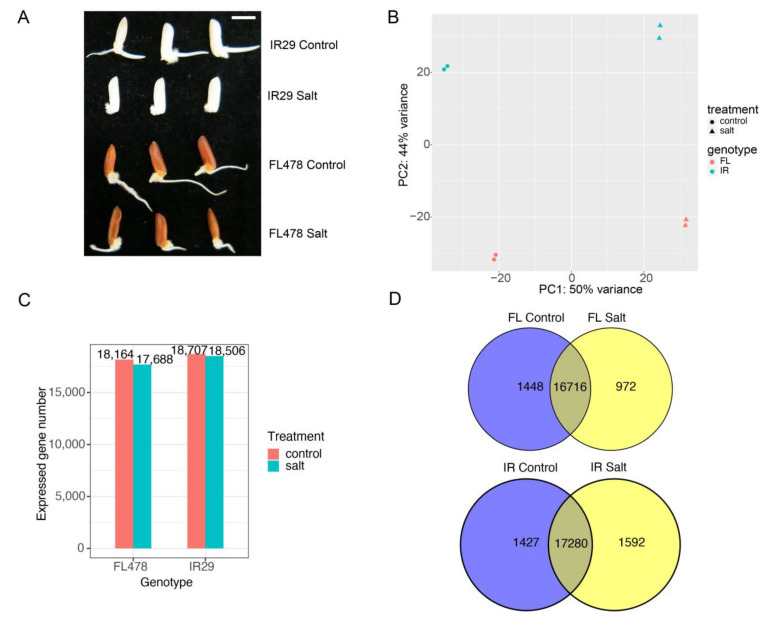
Morphologies and transcriptome characterization of FL478 and IR29 during germination two days after imbibition. (**A**) Morphological appearance of two contrasting genotype seeds under control and 150 mM NaCl condition. Scale bar, 5 mm. (**B**) Principal component analysis of transcriptome data showing the relationships between samples. (**C**) The numbers of expressed genes in two genotypes. (**D**) Venn diagram showing the comparison of expressed genes in two contrasting genotypes.

**Figure 2 ijms-24-03978-f002:**
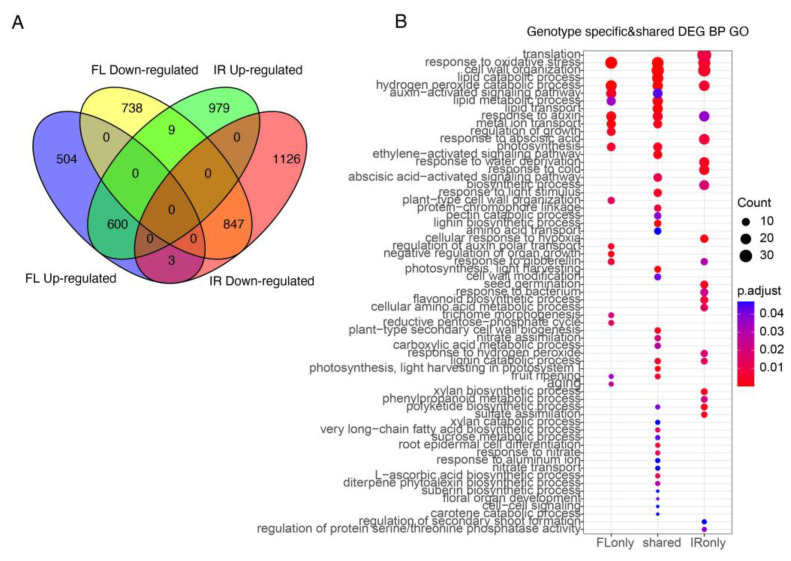
Differentially expressed genes in two genotypes. (**A**) Comparison of upregulated and downregulated genes. (**B**) GO enrichment analysis (biological process) of DEGs.

**Figure 3 ijms-24-03978-f003:**
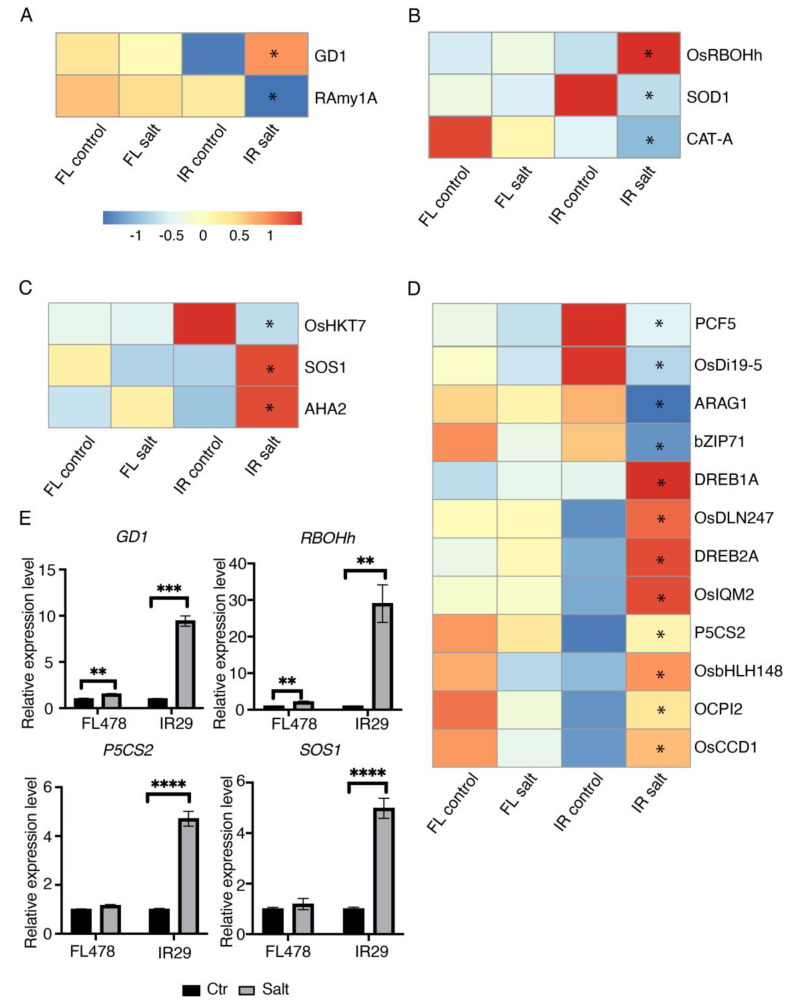
Expression of representative differentially expressed genes. (**A**) GD1 and RAmy1A expression pattern in two genotypes. Z-score normalized. (**B**) The expression pattern of differentially expressed ROS production and scavenging enzymes. (**C**) The expression pattern of ion homeostasis-related DEGs. (**D**) The expression pattern of salt-responsive DEGs. (**E**) qRT-PCR results of GD1, RBOHh, P5CS2, and SOS1 (student‘s *t*-test, * *p* < 0.05, ** *p* < 0.01, *** *p* < 0.001, **** *p* < 0.0001).

**Figure 4 ijms-24-03978-f004:**
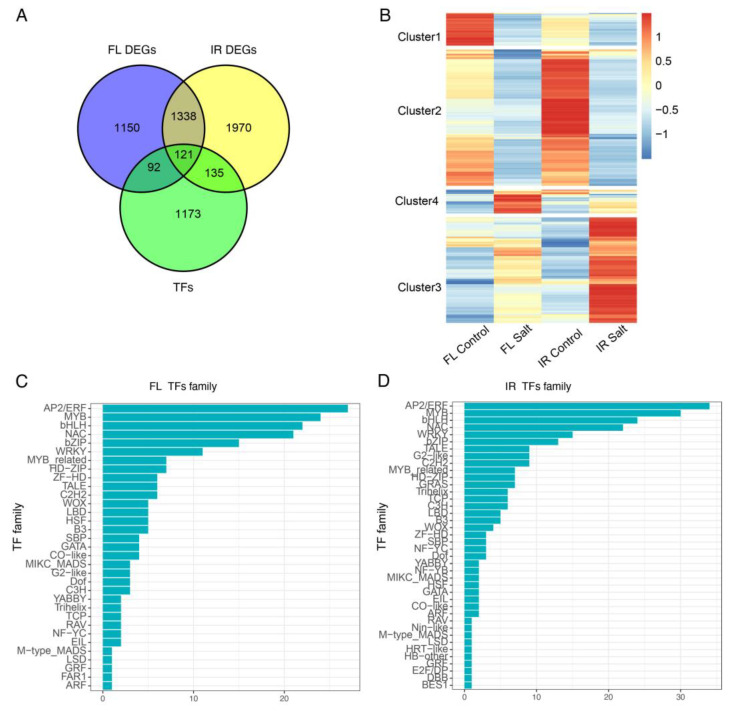
Characterization of differentially expressed salt-responsive transcription factors (TFs) during gemination under salt stress. (**A**) Venn diagram showing the number of salt-responsive TFs in FL478 and IR29. (**B**) Heatmap of differentially expressed TFs, z-score normalized. Differentially expressed TFs families in FL478 (**C**) and IR29 (**D**).

**Figure 5 ijms-24-03978-f005:**
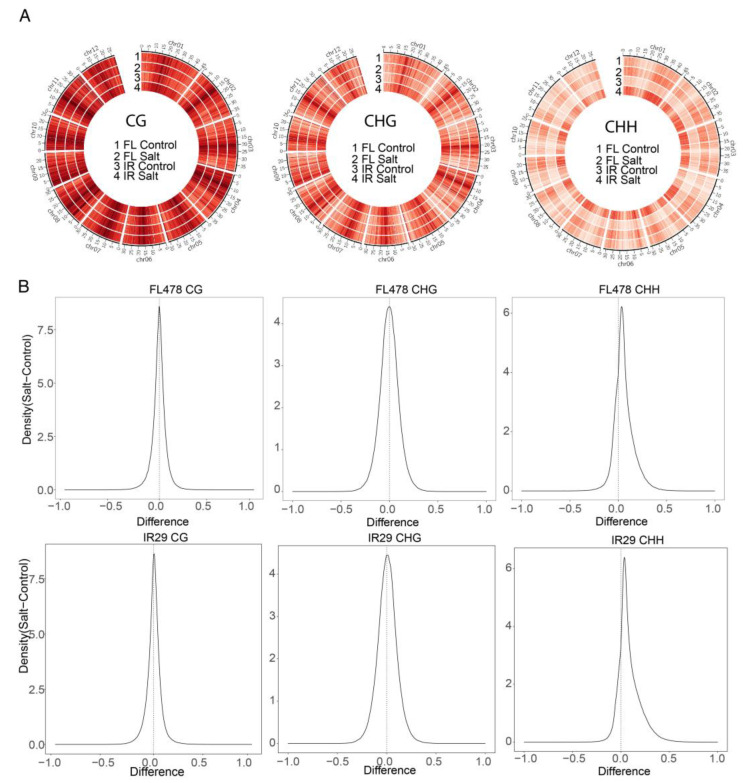
CHH methylation is kept globally high under salt stress. (**A**) whole genome DNA methylation level in two contrasting genotypes in control and saline condition (**B**) Density plot of DNA methylation level difference between salt and control.

**Figure 6 ijms-24-03978-f006:**
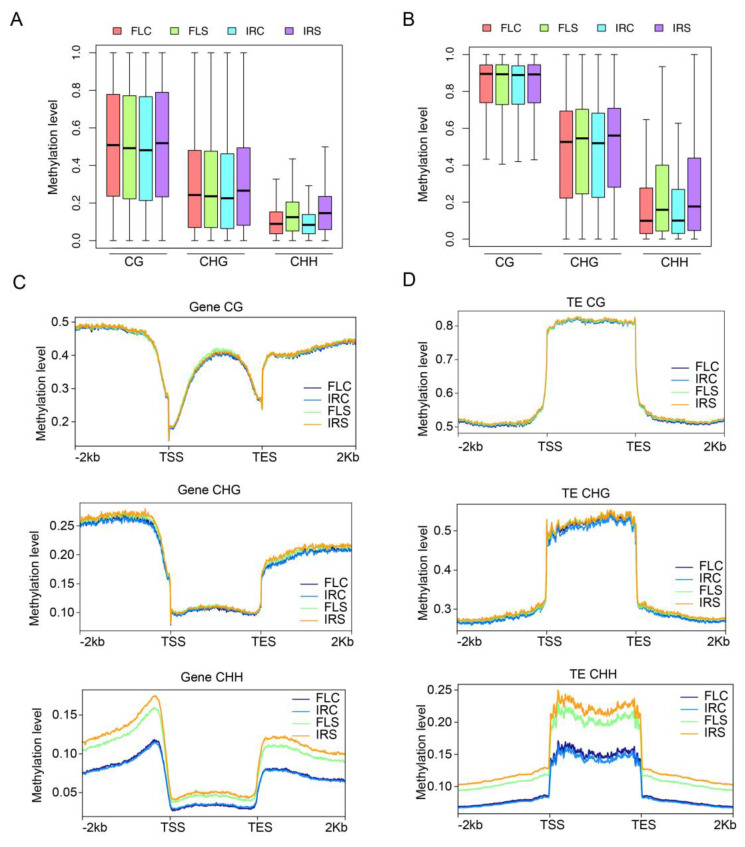
DNA methylation of gene and transposable elements. (**A**) DNA methylation at the promoters of protein-coding genes. (**B**) DNA methylation on TE body. (**C**) DNA methylation profile of gene body, upstream 2 kb, and downstream 2 kb. (**D**) DNA methylation profile of TE body, upstream 2 kb, and downstream 2 kb.

**Figure 7 ijms-24-03978-f007:**
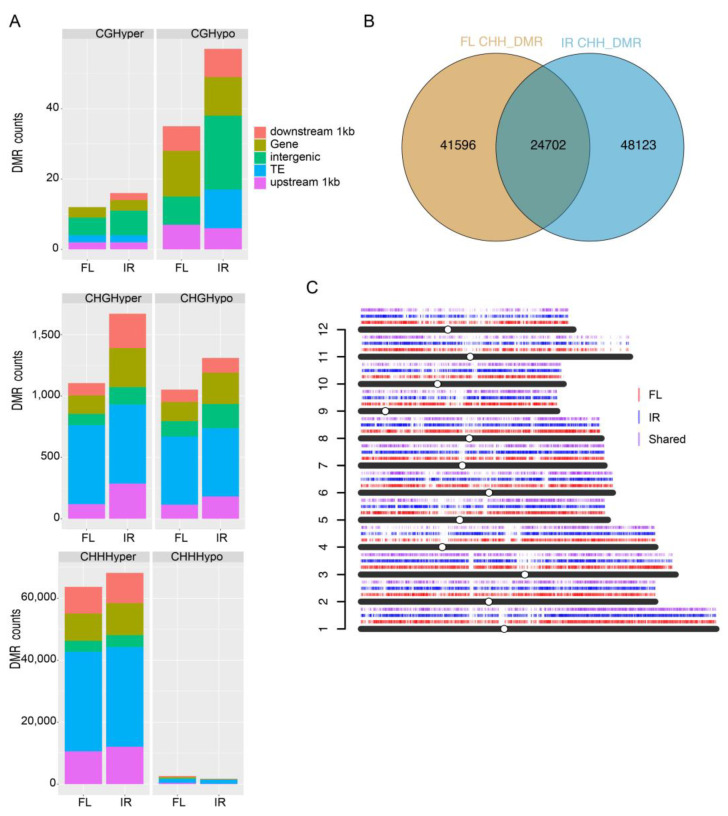
Differentially methylated regions. (**A**) DMR numbers in two genotypes in three contexts and DMR annotation in the genome. (**B**) CHH DMRs comparison in two genotypes. (**C**) CHH DMRs distribution on chromosomes.

**Figure 8 ijms-24-03978-f008:**
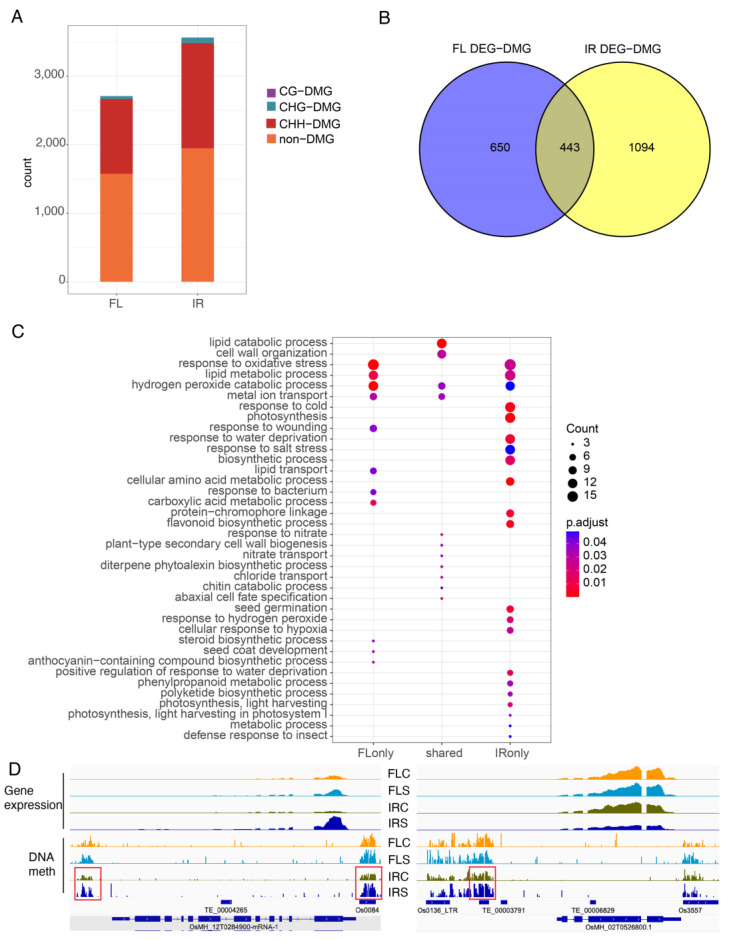
Differentially expressed genes with DMRs (DMG). (**A**) Differentially expressed gene numbers with DMRs or without DMRs. (**B**) Comparison of DMGs in two genotypes. (**C**) GO enrichment analysis of DMGs. (**D**) Two representative DMGs. Red frame indicates DMR. Top panel: Gene expression. Bottom panel: DNA methylation.

## Data Availability

The RNA-seq and BS-seq raw data have been submitted to the National Genomics Data Center (NGDC, https://ngdc.cncb.ac.cn, accessed on 27 November 2022), accession number: PRJCA013516.

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
