# Peer review of "Transcriptome and DNA Methylome Analysis of Two Contrasting Rice Genotypes under Salt Stress during Germination"

_ijms, 2023, doi:10.3390/ijms24043978_

Round 1
Reviewer 1 Report
ABSTRACT
I would recommend to replace the abbreviations DRMs, TE, and GO by their explanation. These abbreviations were used only one time in the abstract.
INTRODUCTION
p.1, line 34 Please cite the GSS map.
p.1, line 52 Please rephrase " Transcriptome and metabolome have been performed in an indica and a japonica variety".
p.1, lines 55-57 Please rephrase the sentence. The mention of NaCl concentration is not necessary.
p.1, lines 50-51 Please clarify the phytohormones abbreviation for the first appearance.
p.1 The references [23] and [18] does not seem to be relevant. I would recommend to focus on the salt stress-associated references. Please consider removal.
RESULTS
It is not necessary to describe the procedures in the section Results. The morphological changes deserve deeper description.
Fig.1, 4 The figure caption should be self-explanatory. Picture 1A should be enriched by scale bar.
Subsection 2.4 I think that it is early to conclude the accumulation of ROS and ion homeostasis based on the mRNA expression levels, without any ROS detection assays. I recommend to modify the title of this subsection. Moreover, the expression of genes identified as up/downregulated should be validated by at least qRT-PCR.
2.6 Please modify the title, I recommend not to use verbs in the titles.
2.7 The first sentence should fit better into Discussion.
p.8, line 232 Please clarify MITE.
2.8 The criteria should be described in the section Material and methods.
MATERIALS AND METHODS
4.1 The authors could describe the experimental groups in more detail. There is no mention about control in this subsection. Did the authors perform any germination tests? How many seeds were included in one replicate?
p.13, line 382 Did the authors mean NOVOGENE Co., Ltd.?
CONCLUSION
I would recommend to enhance the conclusion by short future perspectives of your results.
Author Response
Dear Reviewer,
We appreciate you for your precious time in reviewing our paper and providing valuable points. It was your valuable and insightful points that led to possible improvements in the current version. The authors have carefully considered the points and tried our best to address every one of them. We hope the manuscript after careful revisions meet your high standards. The authors welcome further constructive points if any.
Below we provide the point-by-point responses. All revisions in the manuscript have been highlighted in red.
Please see the attachment.

Reviewer 2 Report
IN L-23-25 what about FL478?
If there are no changes in tolerant cultivar, how it can be used as a QTL market. If it is to be used as marker, then t shouldbe differentially expressed to stress.
Upregulation of a trait is useful feature that could provide salt tolerance. More explanation is needed
L16-17; Fig. 1 972 genes are upregulated due to salt, I guess they are more worthy to be explored
L127 section 2.3: Why GD1 was chosen?
Why not to choose a Tf which is differentially expressed in tolerant as well as susceptible cultivar. TF of choice should be the one which respond to the stress and should being about the change in tolerance. If the aim is to choose for salt tolerance, Tf which is overexpressed under salinity in tolerant cultivar should be chosen
L136-144: Do you believe that only SOD1 and CATa is responsible for ROS alleviation. Other enzymatic, non enzymatic antioxidants and their isoforms should also be studied.
I suggest to highlight the genes which are upregulated in tolerant cultivar under high salinity, and a corresponding decline in susceptible cultivar, to achieve the aim stated in this paper
Author Response

(The authors gave the same response as above.)

Round 2
Reviewer 1 Report
Suggested changes were incorporated into the manuscript, the unsupported conclusions were omitted. I would remove the very last sentence from the Conclusions.